# Perceptual Piercing: Human Visual Cue-based Object Detection in Low Visibility Conditions

## Abstract

This study proposes a novel deep learning framework inspired by atmospheric scattering and human visual cortex mechanisms to enhance object detection under poor visibility scenarios such as fog, smoke, and haze. These conditions pose significant challenges for object recognition, impacting various sectors, including autonomous driving, aviation management, and security systems. The objective is to enhance the precision and reliability of detection systems under adverse environmental conditions. The research investigates the integration of human-like visual cues, particularly focusing on selective attention and environmental adaptability, to ascertain their impact on object detection's computational efficiency and accuracy. This paper proposes a multi-tiered strategy that integrates an initial quick detection process, followed by targeted region-specific dehazing, and concludes with an in-depth detection phase. The approach is validated using the Foggy Cityscapes, RESIDE-$\beta$ (OTS and RTTS) datasets and is anticipated to set new performance standards in detection accuracy while significantly optimizing computational efficiency. The findings offer a viable solution for enhancing object detection in poor visibility and contribute to the broader understanding of integrating human visual principles into deep learning algorithms for intricate visual recognition challenges. The code for perceptual piercing is available here.

## 1 Introduction

Low-visibility conditions such as rain, snow, fog, smoke, or haze present significant challenges in various fields of computer vision and deep learning, such as autonomous vehicles, security and surveillance, maritime navigation, and agricultural robotics. The objective is to develop a deep-learning framework capable of recognizing objects using human visual cues under adverse visibility conditions. The motivation behind this project lies in addressing the substantial difficulties of identifying objects in low-visibility environments, a critical factor in enhancing airport operations during adverse weather.

Poor visibility often leads to aircraft delays, as planes face challenges in taxiing to their gates without clear visual guidance. This situation necessitates more ground support personnel to assist planes in docking, but due to limited ground staff availability, a bottleneck can occur, impeding the handling of multiple aircraft and resulting in further delays. These delays can escalate, potentially leading to flight cancellations. Although the initial motivation for this project is rooted in reducing delays in airport operations, the scope of the proposed machine learning model extends beyond airport scenarios to include a broad range of low-visibility environments. The following methods have been proposed:

- **Selective Region Enhancement:** Unlike uniform dehazing, focusing on specific regions can reduce processing time and prevent image quality degradation in areas where clarity might introduce false positives or where detail is not essential for current detection goals.

- **Integration with Object Detection:** By bridging the gap between image enhancement and object detection, we offer a cohesive approach that leverages the strengths of both methodologies, addressing the limitations of traditional, separate systems.

The above contributions are inspired by mechanisms of the human visual system, including selective attention, foveal and peripheral vision, human-eye adjustments to environmental conditions,

eye-tracking concepts, bottom-up signals based on sensory input, and top-down processes guided by priors and current goals.

The rest of the article is organized as follows: Section 2 reviews the related work, outlining previous studies and developments pertinent to object detection in low-visibility conditions and the integration of human visual cues into machine learning models. This section also highlights the gaps in current research that our study aims to address. Section 3 describes the methodology of our study, detailing the proposed deep-learning framework inspired by human visual signals, the selection criteria for our datasets, and the experimental setup used to evaluate the model's performance under various low-visibility scenarios.

## 2 RELATED WORK

The field of navigation and detection in low-visibility conditions has seen significant advancements through various methodologies including sensor fusion, visual cue integration, and computational techniques. Aircraft landing has been a focus area, with studies exploring sensor fusion of visible and virtual imagery (Liu et al., 2014) and visual-inertial navigation algorithms relying on synthetic and real runway features (Zhang et al., 2018). For GPS-denied environments, multi-sensor fusion algorithms have been developed for reliable odometry estimation (Khattak et al., 2019).

Research has also addressed depth visualization for navigation and obstacle avoidance in low-vision scenarios (Lieby et al., 2011). Synthetic Vision Systems and full-windshield Head-Up Displays have been explored to aid drivers and pilots in low visibility (Kramer et al., 2014; Charissis & Papanastasiou, 2010). Novel image enhancement methods for low-light conditions have been proposed (Atom et al., 2020), as well as combinations of visual cues with standard wireless communication for road safety (Boban et al., 2012). The importance of geometrical shapes and colors in Head-Up Displays for driving perception has been emphasized (Zhan et al., 2023).

These advancements, however, face common challenges. These include increased computational complexity due to sophisticated algorithms (Zhang et al., 2018; Atom et al., 2020; Tang et al., 2022), durability and performance issues under variable or extreme environmental conditions (Khattak et al., 2019; Boban et al., 2012), potential over-fitting problems due to limited datasets (Zhang et al., 2018; Khattak et al., 2019), and the need for extensive real-world testing (Liu et al., 2014; Boban et al., 2012; Tang et al., 2022). Some studies also lack clarity in explanations or comprehensive validation (Kramer et al., 2014; Zhan et al., 2023).

In the realm of visual recognition and object detection, researchers have explored integrating human-like processing mechanisms with computational models. Studies have delved into brain mechanisms for object recognition, emphasizing hierarchical, feedforward processes (DiCarlo et al., 2012). Comparisons between human visual processing and deep neural networks (DNNs) have noted human superiority in handling visual distortions and differences in attention mechanisms (Dodge & Karam, 2017; van Dyck et al., 2021). Attempts to direct DNNs' visual attention using human eye-tracking data have shown limited success in mimicking human attention patterns (van Dyck et al., 2022).

Innovative approaches include adversarial learning to enhance feature discrimination and match feature priors (Yang et al., 2023a), biologically inspired models integrating top-down and bottom-up processes for robust visual recognition in robotics (Malowany & Guterman, 2020), and models mimicking the mammalian retina to enhance dehazing capabilities (Zhang et al., 2015). Some researchers have proposed models using foveal-peripheral dynamics to reduce computational demands while maintaining high-resolution perception in focused areas (Lukanov et al., 2021).

Recent studies have addressed specific challenges in low-visibility conditions such as fog, low light, and sandstorms. The YOLOv5s FMG algorithm has been introduced for small target detection, integrating various modules for better accuracy and localization (Zheng et al., 2023). Networks improving image clarity in hazy and sandstorm conditions have been developed using novel MLP-based modules for pixel reconstruction (Gao et al., 2023). The Prior Knowledge-Guided Adversarial Learning (PKAL) approach leverages adversarial learning and feature priors for robust visual recognition under adverse visibility (Yang et al., 2023b).

Enhancements to existing models, such as YOLOv8, have incorporated deformable convolutions and attention mechanisms for better pedestrian and vehicle detection in poor visibility (Wu & Gao, 2023). Comprehensive reviews of image de-hazing techniques have highlighted limitations of non-

learning and meta-heuristic methods in real-time applications (V et al., 2023). The impact of low-level vision techniques on high-level visual recognition tasks has been evaluated, suggesting a more integrated approach for better outcomes in poor visibility conditions (Yang et al., 2020).

Novel techniques like spatiotemporal attention detection have been introduced to discern region-level attention in video sequences (Zhai & Shah, 2006). The Parallel Detecting and Enhancing Models (PDE) framework aims to simultaneously improve object detection and image enhancement (Li et al., 2022). Research in visual saliency detection has explored integrating spatial position priors with background cues (Jian et al., 2021), while studies on early visual cues have examined their role in detecting object boundaries in natural scenes (Mély et al., 2016).

Despite the advancements in the field, several limitations persist across existing studies. Many approaches still struggle with joint optimization of object detection and image enhancement, the detection of out-of-focus and low-contrast objects, and maintaining performance in dynamically changing visibility conditions. This paper addresses these challenges by proposing a methodology that combines human visual cues with computational models for object detection in low-visibility conditions. By leveraging insights from human perception, such as attention mechanisms and contextual understanding, the proposed approach aims to enhance the robustness and accuracy of object detection systems. This integration not only helps in effectively handling varying degrees of visibility but also reduces computational complexity by focusing processing power on areas of interest, similar to human selective attention.

Current techniques often struggle with the computational burden of processing high-resolution images in their entirety and may lack robustness in dynamically changing visibility conditions. Additionally, uniform dehazing techniques attempt to improve visibility across the entire image, which can unnecessarily process visually clear regions. This leads to increased computational load and potential degradation of image parts where high clarity is not essential. Our methodology addresses these issues by focusing processing power on selected regions, reducing unnecessary computations. Furthermore, by adapting the processing intensity based on real-time feedback, we enhance system responsiveness and accuracy under diverse operational conditions.

The proposed approach stands out due to its unique integration of human visual cues into the object detection process, particularly in low-visibility conditions. Unlike existing methods that may not fully optimize computational resources or adapt to varying environmental conditions effectively, the proposed architecture mimics the human eye's capability to focus on relevant areas dynamically. This method not only enhances detection accuracy but also improves computational efficiency by prioritizing resource allocation, which is crucial for real-time applications. By addressing these key aspects, this research aims to push the boundaries of object detection in low-visibility conditions, offering a more robust, efficient, and adaptable solution compared to existing methods.

## 3 METHODOLOGY

The proposed methodology in Figure 1 focuses on developing a novel deep-learning framework inspired by the atmospheric scattering model and the human visual cortex to enhance object detection in low-visibility conditions. The framework employs adaptive image enhancement techniques integrated with an object detection network to explore different integration strategies. The pipeline initiates with a lightweight object detection model to identify regions of interest, which are subsequently leveraged for spatial attention in the dehazing process, followed by a more robust detection model for refined and comprehensive object detection. This architecture will be evaluated across various configurations using both synthetic and real-world foggy datasets, with performance measured using standard object detection metrics such as mean Average Precision (mAP) and image quality metrics like Structural Similarity Index Measure (SSIM)A.1 and Peak Signal-to-Noise Ratio (PSNR)A.2.

### 3.1 DATASETS

#### 3.1.1 FOGGY CITYSCAPES

The Foggy Cityscapes (Sakaridis et al., 2018) dataset is created to address the problem of semantic foggy scene understanding (SFSU). While there has been extensive research on image dehazing and

semantic scene understanding with clear-weather images, SFSU has received little attention. Due to the difficulty in collecting and annotating foggy images, synthetic fog is added to real images depicting clear-weather outdoor scenes. This synthetic fog generation leverages incomplete depth information to create realistic foggy conditions on images from the Cityscapes dataset, resulting in Foggy Cityscapes with 20,550 images. The training set consists of 2975 images, validated on 500 images, and the test set had 1525 images. The key features of the dataset include:

- **Synthetic Fog Generation**: Real clear-weather images are used, and synthetic fog is added using a complete pipeline that employs the transmission map.
- **Data Utilization**: The dataset can be utilized for supervised learning and semi-supervised learning. We generated a foggy dataset using the synthetic transmission map and then performed supervised learning on the synthetic foggy data.

### 3.1.2 RESIDE-$\beta$

The RESIDE-$\beta$ Outdoor Training Set (OTS) is a comprehensive dataset designed to facilitate research in outdoor image dehazing. It addresses the challenges posed by haze in outdoor scenes, which significantly degrades image quality and affects subsequent tasks such as object detection and semantic segmentation. The dataset includes approximately 72,135 outdoor images with varying degrees of haze, enabling robust training of dehazing algorithms. For testing, we are using the RESIDE-$\beta$ (REalistic Single Image DEhazing) dataset (Li et al., 2019). The subset of RESIDE-$\beta$, Real-Time Testing Set (RTTS) consists of 4,322 real-world hazy images with annotations for object detection. The training set consists of 3000 images, validated on 500 images, and the test set had 1500 images.

### 3.2 HUMAN VISUAL CUES

**Selective Attention and Foveation**: The human eye is not equally sensitive to all parts of the visual field. Central vision, or foveal vision, is highly detailed and used for tasks like reading and identifying objects. Peripheral vision is less detailed and more sensitive to motion. The system scans the entire image (peripheral vision) similar to our preliminary detection phase. This will identify areas for a more detailed analysis (foveal vision), mimicking the human approach of not processing every detail with equal clarity but focusing on areas of interest.

**Adaptation to environmental conditions**: Just as the human visual system adapts to different lighting conditions and levels of visibility (such as adjusting to a dark room after being in bright sunlight), the adaptive dehazing method adjusts the intensity and focus of its processing based on the detection feedback and environmental context, analogous to the way human vision adjusts to ensure optimal perception under varying conditions.

**Eye Tracking and Gaze-directed processing**: Eye-tracking monitors where a person is looking (the gaze) and what draws attention. In visual processing, this is analogous to directing computational resources toward areas of interest, much like the proposed method focuses on dehazing and detailed detection of regions where objects are likely to be present. By analogy, the system pays more attention to certain parts of the image, just as a person would fixate on specific areas within their field of view when searching for something.

**Integration of Bottom-up and Top-down processes**: The human visual system uses both bottom-up signals (from sensory input) and top-down processes (based on knowledge, expectations, and current goals) to interpret scenes. The proposed model initially uses a bottom-up approach (object detection algorithms flagging potential areas of interest) followed by a top-down approach (focusing on dehazing efforts based on three flagged areas and previous learning), mirroring the complex interplay between sensory data and cognitive processes in human vision.

### 3.3 DEHAZING

**Preliminary Detection**: Implement a lightweight, fast object detection algorithm such as YOLOv5s or YOLOv8n to quickly scan the image for potential regions of interest or active regions and flag those patches with a high likelihood of containing objects. The smaller versions of standard YOLO

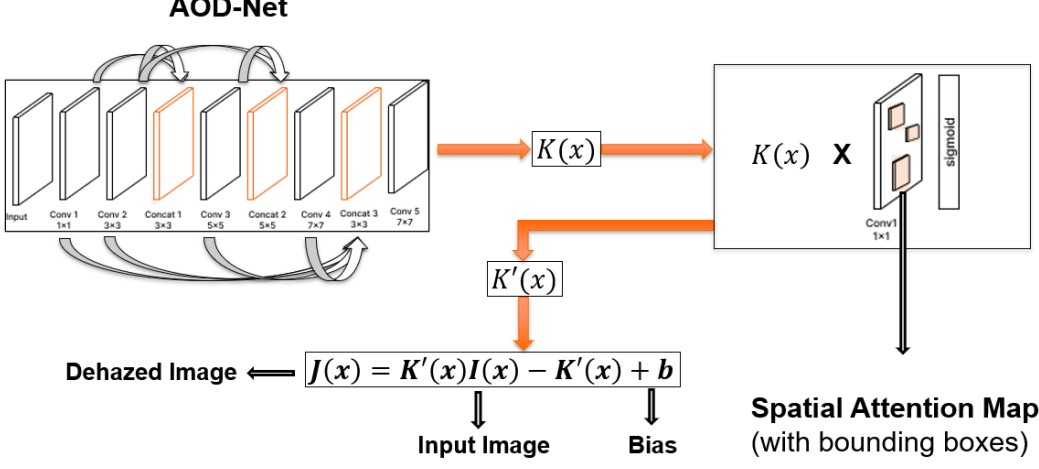

Figure 1: Overall architecture of Perceptual Piercing: (a)Preliminary detection using lightweight object detection model (b) Gaze-directed dehazing using spatial attention on region of interests (c) Final detection using a large and robust model

**AOD-Net**

$$J(x) = K'(x)I(x) - K'(x) + b$$

**Dehazed Image** ⟸

**Spatial Attention Map** (with bounding boxes)

Figure 2: Architecture of AOD-NetX: It takes the transmission map output from the AOD-Net and applies the spatial attention layer to focus on major areas of interest in the given input image.

models are less accurate than their full-sized counterparts but significantly faster, making them ideal for preliminary detection.

**Region-based dehazing**: Apply dehazing algorithms specifically to those active regions identified in the preliminary detection phase. Considering the depth or level of haze, the method should adapt based on the characteristics of the detected regions

The proposed architecture of **AOD-NetX** in Figure 2 utilizes the transmission map created by the standard AOD-Net (Li et al., 2017) and applies it within a spatial attention map module to produce an attention-focused transmission map. This spatial attention map is derived from the bounding boxes or Regions of Interest identified by the lightweight model (YOLOv5s/YOLOv8n) in our proposed method. A sigmoid layer follows, mapping the output probabilities to a range between 0 and 1. We opt not to use softmax in this context due to the independent significance of each bounding box.

### 3.4 OBJECT DETECTION MODELS

The YOLO models used in the detection pipeline include a variety of versions optimized for different purposes. YOLOv5s is a lightweight variant designed for real-time detection with low computational requirements, while YOLOv8n (Nano) is tailored for high-speed applications on resource-constrained devices like mobile phones. On the other hand, YOLOv5x, with its CSP backbone

Table 1: Performance of dehazing methods: AOD-Net and AOD-NetX

| Dataset | Dehazing Method | (Evaluation Metrics) | |
| | | SSIM | PSNR |
| --- | --- | --- | --- |
| **Foggy Cityscapes** | AOD-Net | 0.994 | 26.74 |
| | AOD-NetX | **0.998** | **27.22** |
| **RESIDE-$\beta$ OTS** | AOD-Net | 0.920 | 24.14 |
| | AOD-NetX | **0.945** | **25.80** |
| **RESIDE-$\beta$ RTTS** | AOD-Net | **0.932** | 27.59 |
| | AOD-NetX | 0.656 | **27.62** |

and advanced data augmentation, provides enhanced performance for more complex scenes, and YOLOv8x (Extra Large) offers maximum accuracy for large-scale datasets. The detection process begins by using YOLOv5s or YOLOv8n on foggy images to generate initial annotations. These annotations, along with the original image, are then dehazed using AOD-NetX, and the dehazed image is subsequently passed through YOLOv5x or YOLOv8x for precise and refined detection results.

# 4 RESULTS

The dehazing modules are trained separately on the provided datasets, while the object detection models (various YOLO versions) remain as pre-trained on the MS-COCO dataset. This approach allows users to integrate the dehazing module with their own detection pipeline without requiring a complete re-training of the entire system. However, for improved results, the entire architecture could be fine-tuned on the target datasets, which would serve as a valuable direction for future ablation studies.

## 4.1 DEHAZING PERFORMANCE

The results in Table 1 show that AOD-NetX generally outperforms the standard AOD-Net in terms of SSIM and PSNR across most datasets. For Foggy Cityscapes and RESIDE-$\beta$ OTS, AOD-NetX achieves higher SSIM and PSNR, indicating improved structural similarity and signal quality. However, for RESIDE-$\beta$ RTTS, while AOD-NetX has a slightly better PSNR, AOD-Net achieves a significantly higher SSIM score, suggesting that AOD-Net may retain more structural details in this particular dataset. Overall, AOD-NetX is more effective in most scenarios, especially for complex foggy conditions.

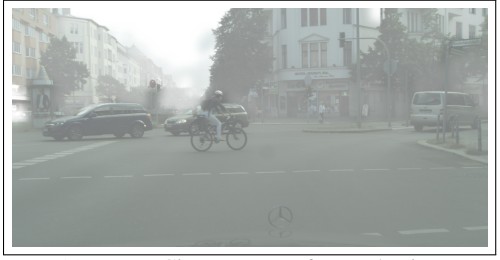
(a) Foggy Cityscapes: Before Dehazing

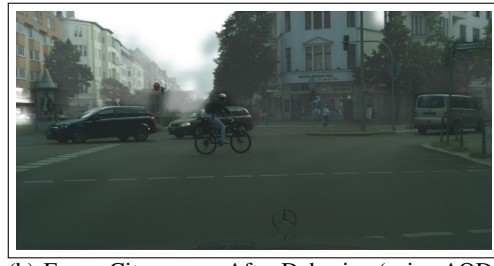
(b) Foggy Cityscapes: After Dehazing (using AOD-NetX)

Figure 3: Dehazing performance on Foggy Cityscapes dataset

Table 2: **Train**- Foggy Cityscapes, **Test**- Foggy Cityscapes: Evaluation of various Perceptual Piercing variations based on mean Average Precision (mAP) under both clear and foggy conditions.

| Architecture Variants | Conditions | Evaluation Metrics (mAP) |
|---|---|---|
| **YOLOv5x** | Clear | 0.5644 |
| | Foggy | 0.485 |
| **AOD-Net+YOLOv5x** | Clear | 0.6813 |
| | Foggy | 0.5822 |
| **YOLOv5s+AOD-NetX+YOLOv5x** | Clear | 0.4896 |
| | Foggy | 0.6152 |
| **YOLOv8x** | Clear | 0.5243 |
| | Foggy | 0.4948 |
| **AOD-Net+YOLOv8x** | Clear | 0.6099 |
| | Foggy | 0.5900 |
| **YOLOv8n+AOD-NetX+YOLOv8x** | Clear | 0.5150 |
| | Foggy | 0.6114 |

## 4.2 PERFORMANCE OF PERCEPTUAL PIERCING

The evaluation results of Perceptual Piercing variations in Table 2 trained and tested on the Foggy Cityscapes dataset indicate that integrating dehazing modules, such as AOD-Net and AOD-NetX, consistently improves object detection performance in both clear and foggy conditions. The 'AOD-Net + YOLOv5x' variant achieved the highest mAP under clear conditions (0.6813), while 'YOLOv5s + AOD-NetX + YOLOv5x' and 'YOLOv8n + AOD-NetX + YOLOv8x' demonstrated the best performance in foggy scenarios, with mAP scores of 0.6152 and 0.6114, respectively. In comparison, baseline YOLO models (YOLOv5x and YOLOv8x) showed lower detection accuracy, highlighting the significance of using enhanced dehazing techniques for better object detection in low-visibility environments.

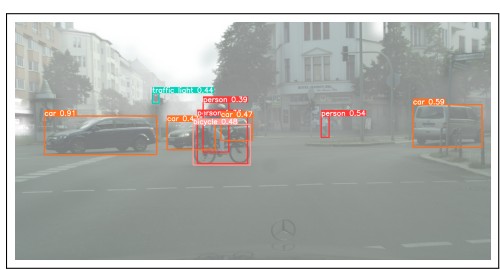 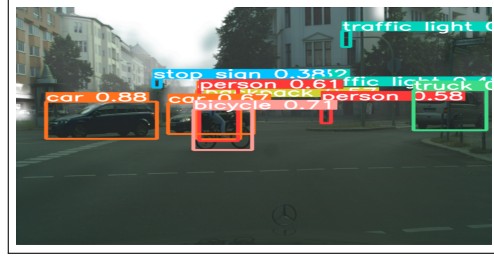

(a) Foggy Cityscapes: Before Dehazing
(b) Foggy Cityscapes: After Dehazing (using AOD-NetX)

Figure 4: Dehazing performance on Foggy Cityscapes dataset

## 4.3 OUT-OF-DISTRIBUTION PERFORMANCE OF PERCEPTUAL PIERCING

The evaluation of various Perceptual Piercing variations in Table 3 trained on Foggy Cityscapes and tested on RESIDE-$\beta$ OTS and RTTS datasets shows that the YOLOv8x architecture achieved the highest mAP scores under foggy conditions, with 0.7125 on OTS and 0.6978 on RTTS. Among the YOLOv5 variants, the baseline YOLOv5x model performed the best, with 0.6944 on OTS and

Table 3: **Train**- Foggy Cityscapes, **Test**- RESIDE-$\beta$ OTS and RTTS: Evaluation of various Perceptual Piercing variations based on mean Average Precision (mAP) under foggy conditions.

| Architecture Variants | Configuration | Evaluation Metrics (mAP) |
|---|---|---|
| **YOLOv5x** | Test: OTS | 0.6944 |
| | Test: RTTS | 0.6655 |
| **AOD-Net+YOLOv5x** | Test: OTS | 0.6325 |
| | Test: RTTS | 0.6156 |
| **YOLOv5s+AOD-NetX+YOLOv5x** | Test: OTS | 0.5679 |
| | Test: RTTS | 0.5297 |
| **YOLOv8x** | Test: OTS | 0.7125 |
| | Test: RTTS | 0.6978 |
| **AOD-Net+YOLOv8x** | Test: OTS | 0.6458 |
| | Test: RTTS | 0.6125 |
| **YOLOv8n+AOD-NetX+YOLOv8x** | Test: OTS | 0.5779 |
| | Test: RTTS | 0.5312 |

0.6655 on RTTS. The addition of AOD-Net generally improved performance for YOLOv8 but had a diminishing effect on YOLOv5. Models incorporating AOD-NetX showed lower mAP values across both test datasets, indicating that its integration may need further optimization. Overall, the results suggest that YOLOv8x is more robust for foggy conditions compared to other variations.

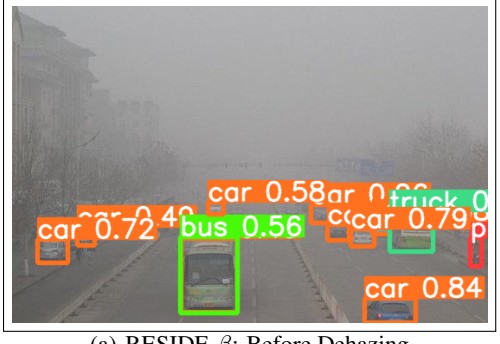
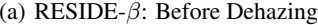
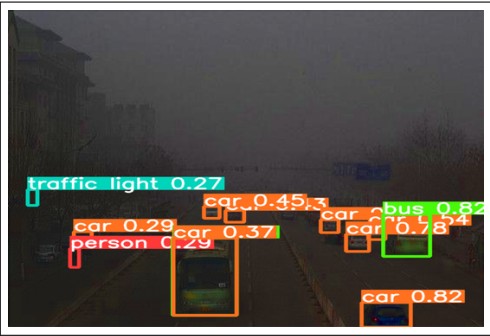

(a) RESIDE-$\beta$: Before Dehazing      (b) RESIDE-$\beta$: After Dehazing (using AOD-NetX)

Figure 5: Dehazing performance on RESIDE-$\beta$ dataset

## 5 DISCUSSION

Integrating a lightweight model with dehazing techniques forms a robust framework that significantly enhances overall system efficiency and effectiveness. This combined approach not only addresses the inherent limitations found in isolated systems but also synergizes their strengths to improve image clarity and object detection accuracy. By adopting a human-vision-inspired architecture, this methodology not only meets but exceeds the performance benchmarks set by state-of-the-art (SOTA) object detection models when tested against the same dataset distribution.

Furthermore, our directed dehazing strategy, which systematically targets specific image impairments, yields superior results with considerably fewer computations compared to traditional dehaz-

ing methods. This efficiency is pivotal, especially in real-time applications where computational resources and response times are critical factors. The success of this approach illustrates the potential of leveraging domain-specific enhancements to refine the capabilities of general object detection frameworks, suggesting a promising direction for future research and development in image processing technologies.

## 5.1 LIMITATIONS

The primary limitations of this paper are as follows: First, the proposed bio-inspired architecture does not incorporate image understanding from various low-visibility scenarios, which could have provided a more comprehensive validation of the methodology. Second, the scope of low-visibility images used is limited to foggy conditions, excluding other challenging environments such as rainy or hazy scenes. Extending the evaluation to rain or combined distribution datasets would enhance the robustness of the framework. Third, while the methodology aims for computational efficiency, the two-tiered detection process coupled with intensive region-specific dehazing may still require substantial computational resources, potentially limiting its applicability in real-time scenarios. Finally, in Out-of-Distribution (OOD) testing, the performance degrades compared to a more generalized model (e.g., YOLOv5x or YOLOv8x). It has been observed that even in clear images within the OOD dataset, the performance declines. This occurs because the dehazing model's embedding space predominantly consists of foggy images, making it less effective when applied to clear scenarios in OOD datasets.

## 5.2 FUTURE WORK

To address the issue of generalizability in single-dataset training, two potential approaches are proposed. The first involves selectively applying the dehazing pipeline only when the scene is sufficiently hazy, determined using a simple haze index computed based on image contrast, brightness, and texture. The second approach is to train the dehazing model with embeddings from both foggy and clear images, thereby enabling it to generalize more effectively across diverse visibility conditions. To further enhance the robustness and applicability of our model, future research should focus on expanding testing with additional datasets that encompass a broader spectrum of low-visibility scenarios, including diverse environmental conditions such as rain, snow, and various levels of nighttime darkness. Such enhancements will enable the model to handle a wider range of adverse weather conditions, increasing its versatility and applicability in real-world situations. Moreover, incorporating training on more diverse datasets is crucial for improving generalization and optimizing performance in out-of-distribution testing. The availability of 4K datasets, which allow for the use of bounding box crops in dehazing, presents an opportunity to refine the model's effectiveness further. Future efforts could also explore optimizing the model architecture and employing more advanced computational techniques to reduce resource demands, thereby enhancing feasibility for real-time applications in autonomous vehicles and other critical systems.

## 6 CONCLUSION

In conclusion, our research addresses the challenge of object detection under adverse conditions like fog, smoke, and haze, which commonly impair autonomous driving, aviation, and security. These environmental factors significantly degrade detection system performance, highlighting the need for precise, reliable methodologies. Our method uses a lightweight algorithm to identify regions of interest, followed by targeted dehazing to enhance visibility where needed most. The clarified images are processed through a robust detection model, boosting accuracy. This approach improves system efficiency and reliability for critical applications across various environments.

Our proposed AODNetX architecture outperforms state-of-the-art models, excelling in both standard and out-of-distribution datasets. This achievement aims to set new benchmarks in detection accuracy and efficiency. Moreover, our approach integrates atmospheric scattering model concepts and human visual cortex insights into machine learning frameworks. The expected outcome is an effective enhancement of object detection under challenging visibility, advancing safety and efficiency in technology-dependent sectors. This integration not only advances current detection systems but also deepens our understanding of visual processing in complex scenarios.

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

# A    EVALUATION METRICS

## A.1    STRUCTURAL SIMILARITY INDEX MEASURE (SSIM)

The performance of dehazing methods is evaluated by the following equation of SSIM score between two images:

$$SSIM(x,y) = \frac{(2\mu_x\mu_y + c_1)(2\sigma_{xy} + c_2)}{(\mu_x^2 + \mu_y^2 + c_1)(\sigma_x^2 + \sigma_y^2 + c_2)} \tag{1}$$

where:

- $\mu_x$ and $\mu_y$ are the average of $x$ and $y$ respectively.

- $\sigma_x^2$ and $\sigma_y^2$ are the variance of $x$ and $y$ respectively.

- $\sigma_{xy}$ is the covariance of $x$ and $y$.

- $c_1 = (k_1 L)^2$ and $c_2 = (k_2 L)^2$ are two variables to stabilize the division with weak denominator; $L$ is the dynamic range of the pixel-values (typically this is $2^{bits\_per\_pixel} - 1$), $k_1 = 0.01$ and $k_2 = 0.03$ by default.

## A.2    PEAK SIGNAL-TO-NOISE RATIO (PSNR)

Peak Signal-to-Noise Ratio (PSNR) is a widely used metric for evaluating the quality of reconstructed images or videos compared to the original, reference data. It is expressed in decibels (dB) and is calculated based on the mean squared error (MSE) between the original and the reconstructed images. The formula for PSNR is given by:

$$\text{PSNR} = 10 \cdot \log_{10} \left( \frac{\text{MAX}^2}{\text{MSE}} \right), \tag{2}$$

where MAX is the maximum possible pixel value of the image (for example, 255 for 8-bit images), and MSE is the mean squared error, defined as:

$$\text{MSE} = \frac{1}{mn} \sum_{i=0}^{m-1} \sum_{j=0}^{n-1} \left( I(i,j) - K(i,j) \right)^2, \tag{3}$$

where $I(i,j)$ represents the pixel value at position $(i,j)$ in the original image, and $K(i,j)$ represents the pixel value at the same position in the reconstructed image. Higher PSNR values generally indicate better reconstruction quality, as they imply a lower MSE and thus less distortion. PSNR is particularly useful for comparing the performance of different image processing algorithms in tasks such as image compression, denoising, and super-resolution.

### A.2.1    MEAN AVERAGE PRECISION (MAP)

For object detection performance, we are using mean Average Precision (mAP):

$$AP = \frac{\sum_{k=1}^{n}(P(k) \times \text{rel}(k))}{\text{number of relevant documents}} \tag{4}$$

where:

- $P(k)$ is the precision at cutoff $k$ in the list.

- $\text{rel}(k)$ is an indicator function equaling 1 if the item at rank $k$ is a relevant document, 0 otherwise.

- $n$ is the number of retrieved documents.

The mean Average Precision is then calculated as:

$$mAP = \frac{\sum_{q=1}^{Q} AP_q}{Q} \tag{5}$$

where $AP_q$ is the Average Precision for the $q^{th}$ query and $Q$ is the total number of queries.

