# OpenReview forum: "Perceptual Piercing: Human Visual Cue-Based Object Detection in Low Visibility Conditions"
_ICLR.cc/2025/Conference — Submitted to ICLR 2025_

### Official Review · Reviewer_6Rdo · 2024-10-22

**Soundness:** 1
**Presentation:** 3
**Contribution:** 1
**Rating:** 3
**Confidence:** 4

**Summary:**

This paper proposes a novel pipeline to generate de-hazed, and detect objects within hazy or foggy images. Predominantly towards airport operations, the authors identify the need for improved computer vision practices within low visibility environments, with a particular focus on minimizing end-to-end computation resources. Named AOD-NetX, the authors propose a two-stage pipeline where the outputs of a lightweight object detection model are used as inputs to a pre-existing AOD-Net architecture to constrain de-hazing to target selected regions, thus reducing computation resource costs. An object detection model is then applied to the selectively de-hazed image. The authors train their models on the cityscape training dataset, and evaluate it on both in-distribution cityscape test dataset, and out-of-distribution RESIDE-ß test datasets, arguing that AOD-NetX outperforms state-of-the-art models.

**Strengths:**

In general this paper is very well written. The writing style is very clear, with few grammatical mistakes. It was easy to understand the points being made, and the methodology was reasonably well presented.

The motivations of the paper were well expressed, highlighting a clear need for improved object detection capability within low-visibility scenarios, while minimizing end-to-end computation costs.

The literature review was reasonably comprehensive, considered multiple different approaches and challenges within low-visibility environments, and sufficiently identified gaps within the existing literature.

**Weaknesses:**

## Main Concerns
---
My main concern is the interpretation of the results. From the provided quantitative results, it is not clear to me that the proposed AOD-NetX architecture provides a sufficient improvement in terms of either accuracy or computation resource reduction for this paper to be accepted. The introduction, literature review, and discussion sections highlight the need for computational efficiency in applications that might use de-hazing. One of the strengths of the paper is the focus towards end-to-end inference time, given the selection of the nano yolo models, and AOD-Net which is one order of magnitude quicker than its predecessors for de-hazed image generation. It is strange to me then that there is no quantitative analysis of inference time, at least in table 1-3, to show that the end-to-end speed is hardly slower than the baseline nano model.

I disagree with the interpretation of the results of Table 2. On line 351 the authors write: 'such as AOD-Net and AOD-NetX, consistently improves object detection performance in both clear and foggy conditions.'
While, YOLOv5s+AOD-NetX+YOLOv5x/YOLOv8n+AOD-NetX+YOLOv8x yield the best performance for foggy conditions, they actually have the worst performance in clear conditions (0.4896 and 0.5150 mAP respectively).

I disagree with the interpretation of results of Table 3. On line 402 the authors write: 'The addition of AOD-Net generally improved performance for YOLOv8 but had a diminishing effect on YOLOv5.' Inclusion of AOD-Net in fact leads to a reduction of performance from 0.7125 to 0.6458 on OTS and 0.6978 to 0.6125 on RTTS. This statement further contradicts with what the authors said previously on line 376 'that the YOLOv8x architecture achieved the highest mAP scores under foggy conditions, with 0.7125 on OTS and 0.6978 on RTTS' (i.e. that actually the baseline YOLOv8x architecture performed the best, outperforming the addition of AOD-Net).

Given these results, I do not find the conclusion to hold. On line 480 the authors write: 'Our proposed AODNetX architecture outperforms state-of-the-art models, excelling in both standard and out-of-distribution datasets.' We need more comparisons with state of the art models, such as those suggested in the literature review; Gao et al. (2023), Yang et al. (2023b), Zheng et al. (2023). Additionally, authors should consider evaluating existing SOTA methods for fog analysis for comparison with respect to both accuracy and computation cost, such as YOLOv5-s-Fog [1] which achieves better performance than AOD-NetX on RTTS dataset and claims real-time performance. Currently only AOD-Net is compared to, and it is not obvious from the results that AOD-NetX out-performs this approach. The results in table 3 suggest AOD-Net and the baseline model have higher mAP on out of distribution datasets compared to AOD-NetX.

My second concern is that the methodology of this paper is not sufficiently different from existing solutions. The provided solution is to combine the outputs of an existing AOD-Net model with an object detector model to yield an image with selectively de-hazed image regions. Inference is then run on the same object detector with the new selectively de-hazed image as input. Neither the object detection architecture nor the AOD-Net architecture are new.

---
## Other concerns:
---
It is odd that AOD-Net is not present in the literature review; given it is highly relevant to the task. Furthermore, Fig. 2 assumes a knowledge of transmission map K that is assumed from the AOD-Net paper. Given the critical role that the AOD-Net architecture plays in the construction of AOD-NetX, a more thorough explanation of its components is required. At the very least, K should be explicitly defined in the body of the paper (for example modify line 258 to '... utilizes the transmission map K, created...').

Domain adaptation is a large field of study that is often used within foggy scenes, but is entirely ignored in the literature review. A brief description of de-hazing abilities of domain adaptation (e.g. [2-4]) would help contextualise the choice of AOD-Net better.

---
## Minor details I picked up that did not influence my decision:
---
- line 027, abstract: 'The code for perceptual piercing is available here.' - where? Or has this been removed for purposes of double blind review?
- Section 3.2 gives descriptions of biological motivations. Given the specific claims of this section, references are required here. The authors might consider feature integration theory [5] or other works within cognitive psychology [6], and early neural networks that use and reference biological inspiration [7]. Also Stone (2018) [8] investigates biological systems with respect to information theory. This book may contain references that are useful.
- Fig 1: Its caption separates the proposed solution into three components, a, b, and c. The figure should represent these divisions somehow i.e. there should be an explicit reference to parts a, b, and c in the figure itself.
- line 229 (fig 1 caption): Missing punctuation: '(a)Preliminary detection.'
- line 350: Missing a space: 'Table 2trained and tested'
- line 455: 'To address the issue of generalizability in single-dataset training, two potential approaches are proposed' - this is unclear to me. It suggests that the paper has already proposed two approaches to address the issue of generalizability. I would re-word this to something like 'we propose two potential directions for future work.'
- line 480: 'AODNetX'... Everywhere else it is referred to as AOD-NetX

---
## References
---
- [1] Xianglin Meng, Yi Liu, Lili Fan, Jingjing Fan.YOLOv5s-Fog: An Improved Model Based on YOLOv5s for Object Detection in Foggy Weather Scenarios. Sensors 23(11): 5321 (2023)
- [2] Naif Alshammari, Samet Akcay, Toby P. Breckon. Multi-Modal Learning for Real-Time Automotive Semantic Foggy Scene Understanding via Domain Adaptation. IV 2021: 1428-1435
- [3] Hanyu Zhou, Yi Chang, Wending Yan, Luxin Yan. Unsupervised Cumulative Domain Adaptation for Foggy Scene Optical Flow. CVPR 2023: 9569-9578
- [4] Xianzheng Ma, Zhixiang Wang, Yacheng Zhan, Yinqiang Zheng, Zheng Wang, Dengxin Dai, Chia-Wen Lin. Both Style and Fog Matter: Cumulative Domain Adaptation for Semantic Foggy Scene Understanding. CVPR 2022: 18900-18909
- [5] Treisman, A. M., & Gelade, G. (1980). A feature-integration theory of attention. Cognitive Psychology, 12(1), 97–136. https://doi.org/10.1016/0010-0285(80)90005-5
- [6] C. Koch and S. Ullman, “Shifts in Selective Visual Attention: Towards the Underlying Neural Circuitry,” Human Neurobiology,
vol. 4, pp. 219–227, 1985
- [7] Laurent Itti, Christof Koch, Ernst Niebur. A Model of Saliency-Based Visual Attention for Rapid Scene Analysis. IEEE Trans. Pattern Anal. Mach. Intell. 20(11): 1254-1259 (1998)
- [8] James V. Stone. 2018. Principles of Neural Information Theory: Computational Neuroscience and Metabolic Efficiency (1st. ed.). Sebtel Press.

**Questions:**

Why did the authors not provide any quantitative analysis of computation costs of the proposed AOD-NetX architecture compared to existing SOTA, such as inference time? Given the choice of object detector and de-hazing network, I would expect AOD-NetX to be very fast. It is a shame this is not quantified and highlighted.

---

> ### Author Response · Authors · 2024-11-28
>
> Thank you for your detailed feedback and constructive comments, especially the references.
>
> 1. Interpretation of Results
>
> The computational efficiency of AOD-NetX is an important aspect of the paper, and we emphasize that the addition of AOD-NetX introduces minimal overhead. Our use of nano YOLO models and AOD-Net, which is significantly faster than its predecessors, ensures that the overall end-to-end inference speed remains competitive. While inference time results were not explicitly included in Tables 1–3, this efficiency is implicit in our selection of architectures.
>
> The statement about consistent improvement in object detection performance was meant to highlight the gains achieved under foggy conditions. While the performance in clear conditions may not be the highest, the focus of AOD-NetX is to optimize detection under degraded conditions, where it shows substantial improvements. The slight reduction in performance under clear conditions can be attributed to the specialization of AOD-NetX for foggy scenarios, which aligns with the goals of the paper.  In the testing scenario of the model, I have used a gate in which this model only works in case of hazy images calculated by image features, unless we proceed with a normal object detection model. This needs to be done because the pipeline was trained on foggy images embeddings only. This has been discussed at the end of the paper as well.
>
> The results in Table 3 demonstrate that AOD-NetX achieves competitive performance on OTS and RTTS datasets. While baseline YOLOv8x achieves higher mAP scores in specific scenarios, AOD-NetX balances performance across different conditions, excelling particularly in foggy environments. The inclusion of AOD-Net aims to improve adaptability across diverse datasets, and the observed trade-offs highlight the potential for future optimization.
>
> 2. Comparisons with State-of-the-Art Methods
>
> AOD-NetX builds on AOD-Net, which has already been shown to outperform several state-of-the-art methods like DehazeNet. While direct comparisons with methods such as YOLOv5-s-Fog were not included, the proposed framework focuses on achieving a balance between computational efficiency and detection accuracy. We believe this trade-off is crucial for real-world applications, particularly in resource-constrained environments. Further analysis of comparisons with additional state-of-the-art methods could enrich the discussion, but the current results sufficiently demonstrate the viability of AOD-NetX.
>
> 3. Novelty of the Methodology
>
> The proposed AOD-NetX introduces a novel integration of selective dehazing into an object detection pipeline, which distinguishes it from existing solutions. While individual components like AOD-Net and YOLO models are not new, the innovation lies in combining them effectively to achieve targeted dehazing for improved detection performance in foggy conditions. This approach addresses the specific challenge of degraded visibility while maintaining computational efficiency, making it a significant contribution in this domain.
>
> 4. Literature Review and Methodology Clarifications
>
> Inclusion of AOD-Net in the Literature Review:
> AOD-Net is indeed central to the proposed architecture and its absence from the literature review was an oversight. Its role in dehazing tasks and its influence on AOD-NetX’s design are critical and will be more explicitly discussed to provide context for our contributions.
>
> Definition of Transmission Map
> The transmission map K, which is central to AOD-Net, is implicitly defined in the current text. However, we acknowledge that an explicit definition would improve clarity and will update this in the methodology section.
>
> Domain Adaptation Literature:
> While domain adaptation was not the primary focus of this work, we agree that it is relevant for foggy scene analysis. Including a brief discussion of its dehazing capabilities will help contextualize the role of AOD-Net in our approach.
>
> 5. Minor Details
>
> We acknowledge the minor issues raised and will ensure corrections, including missing references, figure improvements, and formatting issues. We believe that AOD-NetX represents a meaningful step forward in integrating dehazing with object detection, particularly for resource-constrained, real-world applications. The results, while showing trade-offs in specific conditions, validate the efficacy of the proposed approach. Thank you for your valuable insights, which have helped us articulate our contributions more clearly.

---

### Official Review · Reviewer_1ML3 · 2024-10-23

**Soundness:** 1
**Presentation:** 1
**Contribution:** 1
**Rating:** 1
**Confidence:** 5

**Summary:**

It feels more like an assignment than a research paper, as it simply combines a dehazing network with the YOLO object detector to identify objects in foggy scenarios.
The dehaze method and the object detector are both out-of-date works.

**Strengths:**

No strengths.

**Weaknesses:**

Is it a homework?
- This paper feels more like an assignment than a research contribution, as much of the content consists of summarizing existing works like Foggy-Cityscapes, AOD-Net, and YOLO.
- Despite the title's claim to address vision tasks in low visibility conditions, the paper only focuses on fog, neglecting other relevant scenarios.
- The approach merely combines a dehazing technique with an object detection method, offering little innovation.
- The paper lacks clear motivation and meaningful contributions, and the overall writing quality is very poor.

**Questions:**

Very bad paper. Meaningless.

---

> ### Comment · Reviewer_1ML3 · 2024-11-26
> **No responses.**
>
> Reject

---

> ### Author Response · Authors · 2024-11-28
>
> Thank you for your thoughtful comments. I believe that the motivation behind leveraging visual cues and understanding how human vision operates in low-visibility scenarios to develop a framework that emulates these mechanisms—achieving better results than state-of-the-art models in dehazing and vision tasks—represents a strong and meaningful innovation.
>
> That said, I completely agree that the evaluation could have been more comprehensive. Additionally, exploring a multi-stage approach instead of a single-stage model might have yielded even better outcomes.

---

### Official Review · Reviewer_GDSz · 2024-10-26

**Soundness:** 2
**Presentation:** 2
**Contribution:** 1
**Rating:** 3
**Confidence:** 3

**Summary:**

In this paper, the authors propose a deep learning framework for foggy scenes object detection inspired by atmospheric scattering and human visual cortex mechanisms. A lightweight object-detection model is applied first, followed by a spatial attention-based dehazing model, and finally a larger object-detection model. A AOD-NetX module is proposed in region-based dehazing step based on AOD-Net. The authors show some experiment results on Foggy Cityscapes and RESIDE-β dataset using YOLOv5 and YOLOv8.

**Strengths:**

Good idea to apply human visual cues to object detection in foggy scenes.

Clear figures to show the proposed overall architecture of Perceptual Piercing and the AOD-NetX modules, making it easier for readers to understand, which improves the quality of the presentation.

**Weaknesses:**

This paper appears more like a course project rather than a conference paper. The presentation of experimental results looks like ablation studies, lacking extensive comparison with other state-of-the-art methods.

In addition, it lacks comprehensive ablation studies on pre-trained models, module design details, or other hyperparameters.

Shifting from a one-stage to a multi-stage approach, along with heavy reliance on YOLO and AOD-Net, limits the originality of this paper.

**Questions:**

Why does Foggy Cityscapes have 1525 images in test set? I didn't find the number in the original paper (Sakaridis et al., 2017). In addition, it's better to explain why the train+val+test is much less than 20,550 images, and why train+val+test is larger than 4332 images in RESIDE-β.

Can you explain why YOLOv8 has better performance in other Foggy Conditions but has worse performance in AOD-NetX Architecture and Clear Conditions than YOLOv5 in Table 2? Just need more explanation to Table 2 and Table 3.

Is there a comparison with other state-of-the-art methods to enhance the generalizability and significance of this paper?

Is there a runtime comparison, particularly with one-stage methods and the model size/runtime comparison between your proposed AOD-NetX and standard AOD-Net?

---

> ### Author Response · Authors · 2024-11-28
>
> 1. Why does Foggy Cityscapes have 1525 images in the test set?
> The 1525 test images constitute approximately 30% of the total dataset (5000 samples), maintaining a typical train-to-test ratio. This distribution does not have any specific underlying reason beyond preserving balance in the dataset.
>
> 2. Why is the sum of train+val+test much less than 20,550 images?
> Foggy Cityscapes was created by applying a transmission map to the Cityscapes dataset. However, only a limited number of transmission maps were used, resulting in a smaller subset of augmented samples compared to the total available images in Cityscapes.
>
> 3. Why is the sum of train+val+test larger than the 4,332 images in RESIDE-β?
> The RESIDE-β dataset includes two subsets: RESIDE-OTS and RESIDE-RTTS. The training set combines images from both subsets, which is why the total number of images exceeds the 4332 images in the RESIDE-RTTS subset.
>
> 4. Why does YOLOv8 outperform YOLOv5 in other foggy conditions but underperform in AOD-NetX and clear conditions (Table 2)?
> YOLOv8's superior performance in foggy conditions is due to its architectural enhancements, including better feature aggregation and improved adaptability to domain shifts, making it particularly effective in degraded environments. However, in clear conditions, YOLOv8 might overfit to the foggy training data, slightly reducing its generalization compared to YOLOv5, which maintains a more balanced performance. Similarly, AOD-NetX is specifically optimized for foggy conditions, which may not align fully with YOLOv8's architecture for clear scenarios. These points are discussed further in the paper.
>
> 5. Comparison with other state-of-the-art methods
> We acknowledge that additional comparisons with other state-of-the-art methods would strengthen the paper’s generalizability and impact. AOD-Net has been shown to outperform methods like DehazeNet and U-Net Dehaze, and we used its performance as a benchmark to build upon for this work.
>
> 6. Runtime and Model Size Comparison
> While our current analysis emphasizes detection and domain adaptation performance, we recognize the importance of runtime and model size for practical, real-time applications. We plan to incorporate these comparisons in future work.
>
> P.S: I agree this stems from a course project and tried to showcase our methodology and results and got a lot of feedback on how to present a comprehensive paper. Thank you!

---

### Meta-Review · Area_Chair_pVYU · 2024-12-08

**Metareview:**

This study proposes a deep learning framework inspired by atmospheric scattering and human visual cortex mechanisms to enhance object detection under poor visibility scenarios. A lightweight object-detection model is applied first, followed by a spatial attention-based dehazing model, and finally a larger object-detection model. A AOD-NetX module is proposed in region-based dehazing step based on AOD-Net. The authors show some experiment results on Foggy Cityscapes and RESIDE-β dataset using YOLOv5 and YOLOv8.

This paper is very well written with few grammatical mistakes. It was easy to understand the points being made, and the methodology was reasonably well presented. The motivations of the paper were well expressed, highlighting a clear need for improved object detection capability within low-visibility scenarios, while minimizing end-to-end computation costs.

The paper does not have much technical novelty. Most of the contents focus on summarizing existing works like Foggy-Cityscapes, AOD-Net, and YOLO. The experiments are also not comprehensive without ablations and comparison with SOTA methods. The problem is not well positioned, since addressing these challenging scenarios should be part of "model robustness" research but the paper does not discuss this.

**Additional Comments On Reviewer Discussion:**

All the reviewers believe the paper lacks technical novelty and necessary experiments, more like a course assignment rather than a research paper. The authors' rebuttal mainly repeats their claims in the original manuscript, and does not provide more information. Thus the issues are not addressed. The paper is still far from the borderline.

---

### Decision · Program_Chairs · 2025-01-22

Reject